# Certifying Geometric Robustness of Neural Networks

**Mislav Balunović, Maximilian Baader, Gagandeep Singh, Timon Gehr, Martin Vechev**
Department of Computer Science
ETH Zurich
{mislav.balunovic, mbaader, gsingh, timon.gehr, martin.vechev}@inf.ethz.ch

## Abstract

The use of neural networks in safety-critical computer vision systems calls for their robustness *certification* against natural geometric transformations (e.g., rotation, scaling). However, current certification methods target mostly norm-based pixel perturbations and cannot certify robustness against geometric transformations. In this work, we propose a new method to compute sound and asymptotically optimal linear relaxations for any composition of transformations. Our method is based on a novel combination of sampling and optimization. We implemented the method in a system called DEEPG and demonstrated that it certifies significantly more complex geometric transformations than existing methods on both defended and undefended networks while scaling to large architectures.

## 1 Introduction

Robustness against geometric transformations is a critical property that neural networks deployed in computer vision systems should satisfy. However, recent work [1, 2, 3] has shown that by using natural transformations (e.g., rotations), one can generate *adversarial examples* [4, 5] that cause the network to misclassify the image, posing a safety threat to the entire system. To address this issue, one would ideally like to *prove* that a given network is free of such geometric adversarial examples. While there has been substantial work on certifying robustness to changes in pixel intensity (e.g., [6, 7, 8]), only the recent work of [9] proposed a method to certify robustness to geometric transformations. Its key idea is summarized in Fig. 1: Here, the goal is to prove that any image obtained by translating the original image by some $\delta_x, \delta_y \in [-4, 4]$ is classified to label 3. To accomplish this task, [9] propagates the image and the parameters $\delta_x, \delta_y$ through every component of the transformation using interval bound propagation. The output region $I$ is a convex shape capturing all images that can be obtained by translating the original image between $-4$ and $4$ pixels. Finally, $I$ is fed to a standard neural network verifier which tries to prove that all images in $I$ classify to 3. This method can also be improved using tighter relaxation based on Polyhedra [10]. Unfortunately, as we show later, bound propagation is not satisfactory. The core issue is that any approach based on bound propagation inherently accumulates loss for every intermediate result, often producing regions that are too coarse to allow the neural network verifier to succeed. Instead, we propose a new method based on sampling and optimization which computes a convex relaxation for the *entire composition* of transformations.

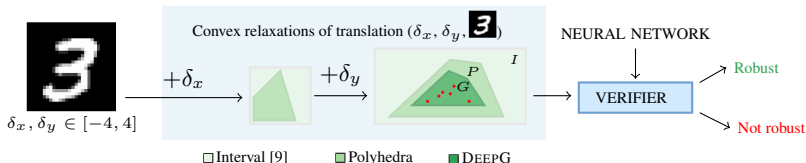

Figure 1: End-to-end certification of geometric robustness using different convex relaxations.

The key idea of our method is to sample the parameters of the transformation (e.g., $\delta_x, \delta_y$), obtaining sampled points at the output (red dots in Fig. 1), and to then compute sound and asymptotically optimal linear constraints around these points (shape $G$). We implemented our method in a system called DEEPG and showed that it is significantly more precise than bound propagation (using Interval or Polyhedra relaxation) on a wide range of geometric transformations. To the best of our knowledge, DEEPG is currently the state-of-the-art system for certifying geometric robustness of neural networks.

**Main contributions**    Our main contributions are:

- A novel method, combining sampling and optimization, to compute asymptotically optimal linear constraints bounding the set of geometrically transformed images. We demonstrate that these constraints enable significantly more precise certification compared to prior work.

- A complete implementation of our certification in a system called DEEPG. Our results show substantial benefits over the state-of-the-art across a range of geometric transformations. We make DEEPG publicly available at `https://github.com/eth-sri/deepg/`.

## 2   Related work

We now discuss some of the closely related work in certification of the neural networks and their robustness to geometric transformations.

**Certification of neural networks**    Recently, a wide range of methods have been proposed to certify robustness of neural networks against adversarial examples. Those methods are typically based on abstract interpretation [6, 7], linear relaxation [8, 11, 12], duality [13], SMT solving [14, 15, 16], mixed integer programming [17], symbolic intervals [18], Lipschitz optimization [19], semi-definite relaxations [20] and combining approximations with solvers [21, 22]. Certification procedures can also be extended into end-to-end training of neural networks to be provably robust against adversarial examples [23, 24]. Recent line of work [25, 26, 27] proposes to construct a classifier based on the smoothed neural network which comes with probabilistic guarantees on the robustness against $L_2$ perturbations. None of these works except [9] consider geometric transformations, while [9] only verifies robustness against rotation. The work of [28] also generates linear relaxations of non-linear specifications, but they do not handle geometric transformations. We remark that [1] considers a much more restricted (discrete) setting leading to a finite number of images. This means that certification can be performed by brute-force enumeration of this finite set of transformed images. In our setting, as we will see, this is not possible, as we are dealing with an uncountable set of transformed images.

**Neural networks and geometric transformations**    There has been considerable research in empirical quantification of geometric robustness of neural networks [2, 3, 29, 30, 31, 32]. Another line of work focuses on the design of architectures which possess an inherent ability to learn to be more robust against such transformations [33, 34]. However, all of these approaches offer only empirical evidence of robustness. Instead, our focus is to provide formal guarantees.

**Certification of geometric transformations**    Prior work [9] introduced a method for analyzing rotations using the interval propagation and performed certification using the state-of-the-art verifier DEEPPOLY. It is straightforward to generalize their interval approach to handle more complex transformations beyond rotation (we provide details in Appendix A.4). However, as we show experimentally, interval propagation loses precision which is why certification often does not succeed.

To capture relationships between pixel values and transformations, one would ideally use the Polyhedra relaxation [10] instead of intervals. While Polyhedra offers higher precision, its worst-case running time is exponential in the number of variables [35]. Hence, it does not scale to geometric transformations, where every pixel introduces a new variable. Thus, we extended the recent DeepPoly relaxation [9] (a restricted Polyhedra) with custom approximations for the operations used in several geometric transformations (e.g., translation, scaling). Our experimental results show that even though this approach significantly improves over intervals, it is not precise enough to certify robustness of most images in our dataset. In turn, this motivates the method introduced in this paper.

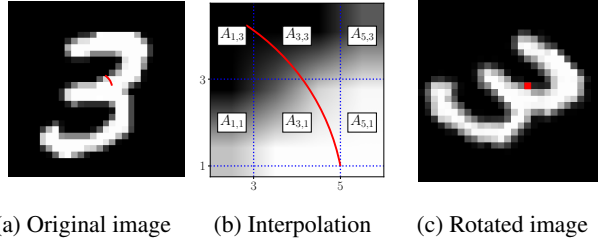

|                    |                  |                    |
|:------------------:|:----------------:|:------------------:|
| (a) Original image | (b) Interpolation | (c) Rotated image  |

Figure 2: Image rotated by $-\frac{\pi}{4}$ degrees. Here, (a) shows the original image, while (b) shows part of (a) with a focus on relevant interpolation regions. Finally, (c) shows the resulting rotated image.

## 3  Background

Our goal is to certify the robustness of a neural network against adversarial examples generated using parameterized geometric transformations. In this section we formulate this problem statement, introduce the notation of transformations and provide a running example which we use throughout the paper to illustrate key concepts.

**Geometric image transformations**  A geometric image transformation consists of a parameterized spatial transformation $\mathcal{T}_{\boldsymbol{\mu}}$, an interpolation $I$ which ensures the result can be represented on a discrete pixel grid, and parameterized changes in brightness and contrast $\mathcal{P}_{\alpha,\beta}$. We assume $\mathcal{T}_{\boldsymbol{\mu}}$ is a composition of bijective transformations such as rotation, translation, shearing and scaling (full descriptions of all transformations are in Appendix A.1). While our approach also applies to other interpolation methods, in this work we focus on the case where $I$ is the commonly-used bilinear interpolation.

To ease presentation, we assume the image (with integer coordinates) consists of an even number of rows and columns, is centered around $(0, 0)$, and its coordinates are odd integers. We note that all results hold in the general case (without the assumption).

**Interpolation**  The bilinear interpolation $I \colon \mathbb{R}^2 \to [0, 1]$ evaluated on a coordinate $(x, y) \in \mathbb{R}^2$ is a polynomial of degree 2 given by

$$I(x,y) := \frac{1}{4} \sum_{\delta_i, \delta_j \in \{0,2\}} p_{i+\delta_i, j+\delta_j} (2 - |i + \delta_i - x|)(2 - |j + \delta_j - y|). \tag{1}$$

Here, $(i, j)$ is the lower-left corner of an *interpolation region* $A_{i,j} := [i, i+2] \times [j, j+2]$ which contains pixel $(x, y)$. Matrix $p$ consists of pixel values at corresponding coordinates in the original image. The function $I$ is continuous on $\mathbb{R}^2$ and smooth on the interior of every interpolation region. These interpolation regions are depicted with the blue horizontal and vertical dotted lines in Fig. 2b.

The pixel value $\tilde{p}_{x,y}$ of the transformed image can be obtained by (i) calculating the preimage of the coordinate $(x, y)$ under $\mathcal{T}_{\boldsymbol{\mu}}$, (ii) interpolating the resulting coordinate using $I$ to obtain a value $\xi$, and (iii) applying the changes in contrast and brightness via $P_{\alpha,\beta}(\xi) = \alpha\xi + \beta$, to obtain the final pixel value $\tilde{p}_{x,y} = \mathcal{I}_{\alpha,\beta,\boldsymbol{\mu}}(x, y)$. These three steps are captured by

$$\mathcal{I}_{\alpha,\beta,\boldsymbol{\mu}}(x, y) := \mathcal{P}_{\alpha,\beta} \circ I \circ \mathcal{T}_{\boldsymbol{\mu}}^{-1}(x, y). \tag{2}$$

**Running example**  To illustrate key concepts introduced throughout the paper, we use the running example of an MNIST image [36] shown in Fig. 2. On this image, we will apply a rotation $R_\phi$ with an angle $\phi$. For our running example, we set $\mathcal{P}_{\alpha,\beta}$ to be the identity.

Consider the pixel $\tilde{p}_{5,1}$ in the transformed image shown in Fig. 2c (the pixel is marked with a red dot). The transformed image is obtained by rotating the original image in Fig. 2a by an angle $\phi = -\frac{\pi}{4}$. This results in the pixel value

$$\tilde{p}_{5,1} = I \circ R_{-\frac{\pi}{4}}^{-1}(5, 1) = I(2\sqrt{2}, 3\sqrt{2}) \approx 0.30$$

Here, the preimage of point $(5, 1)$ under $R_{-\frac{\pi}{4}}$ produces the point $(2\sqrt{2}, 3\sqrt{2})$ with non-integer coordinates. This point belongs to the interpolation region $A_{1,3}$ and by applying $I(2\sqrt{2}, 3\sqrt{2})$ to the original image in Fig. 2a, we obtain the final pixel value $\approx 0.30$ for pixel $(5, 1)$ in the rotated image.

**Neural network certification**   To certify robustness of a neural network with respect to a geometric transformation, we rely on the state-of-the-art verifier DeepPoly [9]. For complex properties such as geometric transformations, the verifier needs to receive a convex relaxation of all possible inputs to the network. If this relaxation is too coarse, the verifier will not be able to certify the property.

**Problem statement**   To guarantee robustness, our goal is to compute a convex relaxation of all possible images obtainable via the transformation $\mathcal{I}_{\alpha,\beta,\boldsymbol{\mu}}$. This relaxation can then be provided as an input to a neural network verifier (e.g., DeepPoly). If the verifier proves that the neural network classification is correct for all images in this relaxation, then geometric robustness is proven.

## 4   Asymptotically optimal linear constraints via optimization and sampling

We now present our method for computing the optimal linear approximation (in terms of volume).

**Motivation**   As mentioned earlier, designing custom transformers for every operation incurs precision loss at every step in the sequence of transformations. Our key insight is to define an optimization problem in a way where its solution is the *optimal* (in terms of volume) lower and upper linear constraint for the entire sequence. To solve this optimization problem, we propose a method based on sampling and linear programming. Our method produces, for every pixel, asymptotically optimal lower and upper linear constraints for the *entire composition* of transformations (including interpolation). Such an optimization problem is generally difficult to solve, however, we find that with geometric transformations, our approach is scalable and contributes only a small portion to the entire end-to-end certification running time.

**Optimization problem**   To compute linear constraints for every pixel value, we split the hyperrectangle $h$ representing the set of possible parameters into $s$ splits $\{h_k\}_{k \in [s]}$. Our goal will be to compute *sound* lower and upper linear constraints for the pixel value $\mathcal{I}_{\boldsymbol{\kappa}}(x,y)$ for a given pixel $(x,y)$. Both of these constraints will be linear in the parameters $\boldsymbol{\kappa} = (\alpha, \beta, \boldsymbol{\mu}) \in h_k$. We define *optimal* and *sound* linear (lower and upper) constraints for $\mathcal{I}_{\boldsymbol{\kappa}}(x,y)$ to be a pair of hyperplanes fulfilling

$$\boldsymbol{w}_l^T \boldsymbol{\kappa} + b_l \le \mathcal{I}_{\boldsymbol{\kappa}}(x,y) \quad \forall \boldsymbol{\kappa} \in h_k \tag{3}$$

$$\boldsymbol{w}_u^T \boldsymbol{\kappa} + b_u \ge \mathcal{I}_{\boldsymbol{\kappa}}(x,y) \quad \forall \boldsymbol{\kappa} \in h_k, \tag{4}$$

while minimizing

$$L(\boldsymbol{w}_l, b_l) = \frac{1}{V} \int_{\boldsymbol{\kappa} \in h_k} \left( \mathcal{I}_{\boldsymbol{\kappa}}(x,y) - (b_l + \boldsymbol{w}_l^T \boldsymbol{\kappa}) \right) \mathrm{d}\boldsymbol{\kappa} \tag{5}$$

$$U(\boldsymbol{w}_u, b_u) = \frac{1}{V} \int_{\boldsymbol{\kappa} \in h_k} \left( (b_u + \boldsymbol{w}_u^T \boldsymbol{\kappa}) - \mathcal{I}_{\boldsymbol{\kappa}}(x,y) \right) \mathrm{d}\boldsymbol{\kappa}. \tag{6}$$

Here $V$ denotes the normalization constant equal to the volume of $h_k$. Intuitively, the optimal constraints should result in a convex relaxation of minimum volume. This formulation also allows independent computation for every pixel, facilitating parallelization across pixels. Next, we describe how we obtain lower constraints (upper constraints are computed analogously).

**Step 1: Compute a potentially unsound constraint**   To generate a reasonable but a potentially unsound linear constraint, we sample parameters $\boldsymbol{\kappa}_1, \ldots, \boldsymbol{\kappa}_N$ from $h_k$, approximate the integral in Eq. (5) by its Monte Carlo estimate $L_N$ and enforce the constraints only at the sampled points:

$$\min_{(\boldsymbol{w}_l, b_l) \in W} L_N(\boldsymbol{w}_l, b_l) = \min_{(\boldsymbol{w}_l, b_l) \in W} \frac{1}{N} \sum_{i=1}^{N} \left( \mathcal{I}_{\boldsymbol{\kappa}_i}(x,y) - (b_l + \boldsymbol{w}_l^T \boldsymbol{\kappa}_i) \right),$$

$$b_l + \boldsymbol{w}_l^T \boldsymbol{\kappa}_i \le \mathcal{I}_{\boldsymbol{\kappa}_i}(x,y) \quad \forall i \in [N]. \tag{7}$$

This problem can be solved exactly using linear programming (LP). The solution is a potentially unsound constraint $b_l' + \boldsymbol{w}_l'^T \boldsymbol{\kappa}$ (it may violate the constraint at non-sampled points). For our running example, the region bounded by these potentially unsound lower and upper linear constraints is shown as orange in Fig. 3.

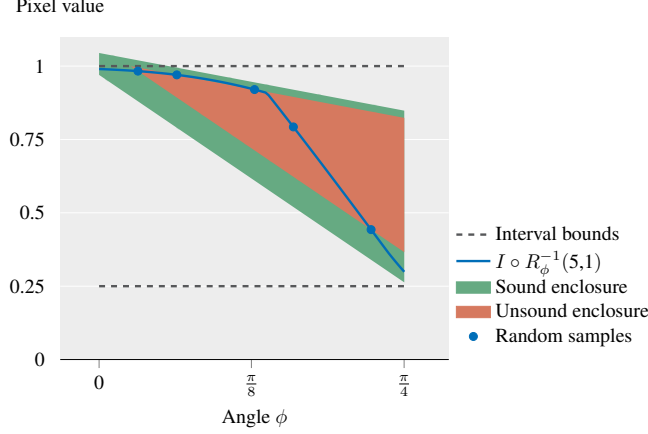

Figure 3: Unsound (Step 1) and sound (Step 3) enclosures for $I \circ R_\phi^{-1}(5, 1)$, with respect to random sampling from $\phi \in [0, \frac{\pi}{4}]$, in comparison to the interval bounds from prior work [9]. Note that $I \circ R_\phi^{-1}(5, 1)$ is not piecewise linear, because bilinear interpolation is a polynomial of degree 2.

**Step 2: Bounding the maximum violation** Our next step is to compute an upper bound on the violation of Eq. (3) induced by our potentially unsound constraint from Step 1. This violation is equal to the maximum of the function $f(\boldsymbol{\kappa}) = b_l' + \boldsymbol{w}_l'^T \boldsymbol{\kappa} - \mathcal{I}_{\boldsymbol{\kappa}}(x, y)$ over the hyperrectangle $h_k$. It can be shown that the function $f$ is Lipschitz continuous which enables application of standard global optimization techniques with guarantees [37]. We remark that such methods have already been applied for optimization over inputs to neural network [38, 19].

We show a high level description of this optimization procedure in Algorithm 1. Throughout the optimization, we maintain a partition of the domain of function $f$ into hyperrectangles $h$. The hyperrectangles are stored in a priority queue $q$ sorted by an upper bound $f_h^{\text{bound}}$ of the maximum value the function can take inside of the hyperrectangle. At every step, shown in Line 6, the hyperrectangle with the highest upper bound is further refined into smaller hyperrectangles $h_1', \ldots, h_k'$ and their upper bounds are recomputed. This procedure finishes when the difference between every upper bound and the maximal value at one of the hyperrectangle centers is at most $\epsilon$. Finally, maximum upper bound of the elements in the queue is returned as a result of the optimization. We provide more details on the optimization algorithm in Appendix A.5.

---

**Algorithm 1** Lipschitz Optimization with Bound Refinement

1: **Input:** $f, h, k \geq 2$
2: $f_{\max} := f(\text{center}(h))$
3: $f_h^{\text{bound}} := f^{\text{bound}}(h, \nabla f)$
4: $q := [(h, f_h^{\text{bound}})]$
5: **repeat**
6:      pop $(h', f_{h'}^{\text{bound}})$ from $q$ with maximum $f_{h'}^{\text{bound}}$
7:      $h_1', \ldots, h_k' := \text{partition}(h', \nabla f)$
8:      **for** $i := 1$ **to** $k$ **do**
9:          $f_{\max} := \max(f(\text{center}(h_i')), f_{\max})$
10:          $f_{h_i'}^{\text{bound}} := f^{\text{bound}}(h_i', \nabla f)$
11:          **if** $f_{h_i'}^{\text{bound}} > f_{\max} + \epsilon$ **then**
12:              add $(h_i', f_{h_i'}^{\text{bound}})$ to $q$
13:          **end if**
14:      **end for**
15: **until** a maximal $f_{h'}^{\text{bound}}$ in $q$ is lower than $f_{\max} + \epsilon$
16: **return** $f_{\max} + \epsilon$

---

The two most important aspects of the algorithm, which determine the speed of convergence, are (i) computation of an upper bound, and (ii) choosing an edge along which to refine the hyperrectangle. To compute an upper bound inside of a hyperrectangle spanned between points $\boldsymbol{h}_l$ and $\boldsymbol{h}_u$, we use:

$$f(\boldsymbol{\kappa}) \le f(\tfrac{\boldsymbol{h}^u + \boldsymbol{h}^l}{2}) + \frac{1}{2} \left| \nabla^{\boldsymbol{h}} f \right|^T (\boldsymbol{h}^u - \boldsymbol{h}^l). \tag{8}$$

Here $\left| \nabla^{\boldsymbol{h}} f \right|$ can be any upper bound on the true gradient which satisfies $|\partial_i f(\boldsymbol{\kappa})| \le \left| \nabla^{\boldsymbol{h}} f \right|_i$ for every dimension $i$. To compute such a bound, we perform reverse-mode automatic differentiation of the function $f$ using interval propagation (this is explained in more details in Appendix A.2). As an added benefit, results of our analysis can be used for pruning of hyperrectangles. We reduce a hyperrectangle to one of its lower-dimensional faces along dimensions for which analysis on gradients proves that the respective partial derivative has a constant sign within the entire hyperrectangle. We also improve on standard refinement heuristics — instead of refining along the largest edge, we additionally weight edge length by an upper bound on the partial derivative of that dimension. In our experiments, we find that these insights speed up convergence compared to simply applying the method out of the box.

Let $v_l$ be the result of the above Lipschitz optimization algorithm. It is then guaranteed that

$$v_l \le \max_{\boldsymbol{\kappa} \in h_k} \left( b'_l + \boldsymbol{w}_l'^T \boldsymbol{\kappa} - \mathcal{I}_{\boldsymbol{\kappa}}(x, y) \right) \le v_l + \epsilon.$$

**Step 3: Compute a sound linear constraint**   In the previous step we obtained a bound on the maximum violation of Eq. (3). Using this bound, in this step, we update our linear constraints $b_l = b'_l - v_l - \epsilon$ and $\boldsymbol{w}_l = \boldsymbol{w}'_l$ to obtain a sound lower linear constraint (it satisfies Eq. (3)). The region bounded by the sound lower and upper linear constraints is shown as green in Fig. 3. It is easy to check that our constraint is sound:

$$b_l + \boldsymbol{w}_l^T \boldsymbol{\kappa} = b'_l - v_l - \epsilon + \boldsymbol{w}_l'^T \boldsymbol{\kappa} \le \mathcal{I}_{\boldsymbol{\kappa}}(x, y) \quad \forall \boldsymbol{\kappa} \in h_k.$$

**Running example**   As in Section 3, we focus on the pixel $(5, 1)$, choose $s = 2$ splits for $[0, \frac{\pi}{2}]$, and focus our attention on the split $[0, \frac{\pi}{4}]$. In Step 1, we sample random points $\{0.1, 0.2, 0.4, 0.5, 0.7\}$ for our parameter $\phi \in [0, \frac{\pi}{4}]$ and evaluate $I \circ R_\phi^{-1}(5, 1)$ on these points, obtaining $\{0.98, 0.97, 0.92, 0.79, 0.44\}$.

These points correspond to the blue dots in Fig. 3. Solving the LP in Eq. (7) yields $b'_l = 1.07$ and $w'_l = -0.90$. Similarly, we compute a potentially unsound upper constraint. Together, these two constraints form the orange enclosure shown in Fig. 3. This enclosure is in fact unsound, as some points on the blue line (those not sampled above) are not included in the region.

In Step 2, using Lipschitz optimization for the function $1.07 - 0.9\phi - I \circ R_\phi^{-1}$ with $\epsilon = 0.02$ over $\phi \in [0, \frac{\pi}{4}]$ we obtain $v_l = 0.08$ resulting in the sound linear lower constraint $0.97 - 0.9\phi$. This, together with the similarly obtained sound upper constraint, forms the sound (green) enclosure in Fig. 3. In the figure we also show the black dashed lines corresponding to the interval bounds from prior work [9] which enclose much larger volume than our linear constraints.

**Asymptotically optimal constraints**   While our constraints may not be optimal, one can show they are asymptotically optimal as we increase the number of samples:

**Theorem 1.** *Let $N$ be the number of points sampled in our algorithm and $\epsilon$ the tolerance used in the Lipschitz optimization. Let $(\boldsymbol{w}_l, b_l)$ be our lower constraint and let $(\boldsymbol{w}^*, b^*)$ be the minimum of L. For every $\delta > 0$ there exists $N_\delta$ such that $|L(\boldsymbol{w}_l, b_l) - L(\boldsymbol{w}^*, b^*)| < \delta + \epsilon$ for every $N > N_\delta$, with high probability. Analogous result holds for upper constraint $(\boldsymbol{w}_u, b_u)$ and function U.*

We provide a proof of Theorem 1 in Appendix A.3. Essentially, as we increase the number of sampled points in our linear program, we approach the optimal constraints. The $\epsilon$ tolerance in Lipschitz optimization could be further decreased towards 0 to obtain an offset as small as desired. In our experiments, we also show empirically that close-to-optimal bounds can be obtained with a relatively small number of samples.

Table 1: Comparison of DEEPG which uses linear constraints with the baseline based on interval bound propagation. Here, R($\phi$) corresponds to rotations with angles between $\pm\phi$; T($x, y$), to translations between $\pm x$ pixels horizontally and between $\pm y$ pixels vertically; Sc($p$), to scaling the image between $\pm p\%$; Sh($m$), to shearing with a shearing factor between $\pm m\%$; and B($\alpha, \beta$), to changes in contrast between $\pm\alpha\%$ and brightness between $\pm\beta$.

| | | Accuracy (%) | Attacked (%) | Certified (%) | |
| --- | --- | --- | --- | --- | --- |
| | | | | Interval [9] | DEEPG |
| MNIST | R(30) | 99.1 | 0.0 | 7.1 | **87.8** |
| | T(2, 2) | 99.1 | 1.0 | 0.0 | **77.0** |
| | Sc(5), R(5), B(5, 0.01) | 99.3 | 0.0 | 0.0 | **34.0** |
| | Sh(2), R(2), Sc(2), B(2, 0.001) | 99.2 | 0.0 | 1.0 | **72.0** |
| Fashion-MNIST | Sc(20) | 91.4 | 11.2 | 19.1 | **70.8** |
| | R(10), B(2, 0.01) | 87.7 | 3.6 | 0.0 | **71.4** |
| | Sc(3), R(3), Sh(2) | 87.2 | 3.5 | 3.5 | **56.6** |
| CIFAR-10 | R(10) | 71.2 | 10.8 | 28.4 | **87.8** |
| | R(2), Sh(2) | 68.5 | 5.6 | 0.0 | **54.2** |
| | Sc(1), R(1), B(1, 0.001) | 73.2 | 3.8 | 0.0 | **54.4** |

## 5    Experimental evaluation

We implemented our certification method in a system called DEEPG. First, we demonstrate that DEEPG can certify robustness to significantly more complex transformations than both prior work and traditional bound propagation approaches based on relational abstractions. Second, we experimentally show that our method requires relatively small number of samples to converge to the optimal linear constraints. Third, we investigate the effectiveness of a variety of training methods to train a network provably robust to geometric transformations. Finally, we demonstrate that DEEPG is scalable and can certify geometric robustness for large networks. We provide networks and code to reproduce the experiments in this paper at `https://github.com/eth-sri/deepg/`.

**Experimental setup**    We evaluate on image recognition datasets: MNIST [36], Fashion-MNIST [39] and CIFAR-10 [40]. For each dataset, we randomly select 100 images from the test set to certify. Among these 100 images, we discard all images that are misclassified even without any transformation. In all experiments for MNIST and Fashion-MNIST we evaluate a 3-layer convolutional neural network with 9 618 neurons, while for the more challenging CIFAR-10 dataset we consider a 4-layer convolutional network with 45 216 neurons. Details of these architectures are provided in Appendix B.2. We certify robustness to composition of transformations such as rotation, translation, scaling, shearing and changes in brightness and contrast. These transformations are formally defined in Appendix A.1. All experiments except the one with large networks were performed on a desktop PC with 2 GeForce RTX 2080 Ti GPU-s and 16-core Intel(R) Core(TM) i9-9900K CPU @ 3.60GHz.

**Comparison with prior work**    In the first set of experiments we certify robustness to geometric transformations and compare our results to prior work [9]. While they considered only rotations, we implemented their approach for other transformations and their compositions. This generalization is described in detail in Appendix A.4 and shown as Interval in Table 1. For each dataset and geometric transformation, we train a neural network using data augmentation with the transformation that we are certifying. Additionally, we use PGD adversarial training to obtain a network robust to noise which we later show significantly increases verifiability of the network. We provide runtime analysis of the experiments and all hyperparameters used for certification in Appendix B.2.

We first measure the success of a randomized attack which samples 100 transformed images uniformly at random [2]. Then, we generate linear constraints using DEEPG, as described in Section 4. Constraints produced by both our method and the interval baseline are given as an input to the state-of-the-art neural network verifier DeepPoly [9]. We invoke both methods for every split separately, with the same set of splits. In order to make results fully comparable, both methods are parallelized over pixels in the same way and the refinement parameter $k$ of interval propagation is chosen such that its runtime is roughly equal to the one of DEEPG. Table 1 shows the results of our evaluation.

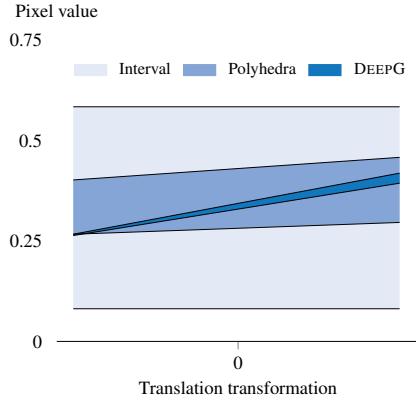

Pixel value

Translation transformation

| | Images certified (%) | | |
|---|---|---|---|
| | Interval | Polyhedra | DeepG |
| T(0.25) | 0 | 14 | **90** |
| Sc(4) | 0 | 23 | **75** |
| Sh(10) | 0 | 12 | **38** |

Figure 4: Translation transformation approximated using interval propagation, polyhedra and DEEPG, for a representative pixel.

Table 2: Certification success rates of interval propagation [9], Polyhedra and DeepG (for translation, shearing and scaling).

While interval propagation used in prior work can prove robustness for simpler transformations, it fails for more complex geometric transformations. For example, already for translation which has two parameters it does not succeed at certifying a single image. This shows that, in order to certify complex transformations, one has to capture relationships between pixel values and transformation parameters using a relational abstraction. Linear constraints computed by DEEPG provide a significant increase in certification precision, justifying the more involved method to compute the constraints.

**Comparison with custom transformers** To understand the benefits of our method further, we decided to construct a more advanced baseline than the interval propagation. In particular, we crafted specialized transformers for DeepPoly [9], which is a restriction of Polyhedra, to handle the operations used in geometric transformations. These kind of transformers have brought significant benefits over the interval propagation in certifying robustness to noise perturbations, and thus, we wanted to see what the benefits would be in our setting of geometric transformations. Concretely, we designed Polyhedra transformers for addition and multiplication which enables handling of geometric operations. These transformers are non-trivial and are listed in Appendix B.1. Fig. 4 shows that relaxation with these transformers is significantly tighter than intervals. This also translates to higher certification rates compared to intervals, shown in Table 2. However, this method still fails to certify many images on which DEEPG succeeds. This experiment shows that generating constraints for the entire composition of transformations as in DEEPG is (expectedly) more effective than crafting transformers for individual operations of the transformation.

**Convergence towards optimal bounds** While Theorem 1 shows that DEEPG obtains optimal linear constraints in the limit, we also experimentally check how quickly our method converges in practice. For this experiment, we consider rotation between $-2°$ and $2°$, composed with scaling between $-5\%$ and $5\%$. We run DEEPG while varying the number of samples used for the LP solver ($n$) and tolerance in Lipschitz optimization ($\epsilon$). In Table 3 we show the approximation error (average distance between lower and upper linear constraint), certification rate and time taken to compute the constraints. For instance, even with only 100 samples and $\epsilon = 0.01$ DEEPG can certify almost every image in 1.2 seconds. While higher number of samples and smaller tolerance are necessary to obtain more precise bounds, they do not bring significant increase in certification rates.

Table 3: Speed of convergence of DEEPG towards optimal linear bounds.

| $n$ | $\epsilon$ | Approximation error | Certified(%) | Runtime(s) |
|---|---|---|---|---|
| 100 | 0.1 | 0.032 | 54.8 | 1.1 |
| 100 | 0.01 | 0.010 | 96.5 | 1.2 |
| 1000 | 0.001 | 0.006 | 97.8 | 4.9 |
| 10000 | 0.00001 | 0.005 | 98.2 | 46.1 |

Table 4: Certification using DEEPG for neural networks trained using different training techniques.

| | | Accuracy (%) | Attack success (%) | Certified (%) | |
|---|---|---|---|---|---|
| | | | | Interval [9] | DEEPG |
| MNIST | Standard | 98.7 | 52.0 | 0.0 | 12.0 |
| | Augmented | 99.0 | 4.0 | 0.0 | 46.5 |
| | $L_\infty$-PGD | 98.9 | 45.5 | 0.0 | 20.2 |
| | $L_\infty$-DIFFAI | 98.4 | 51.0 | 1.0 | 17.0 |
| | $L_\infty$-PGD + Augmented | **99.1** | **1.0** | 0.0 | **77.0** |
| | $L_\infty$-DIFFAI + Augmented | 98.0 | 6.0 | **42.0** | 66.0 |

**Comparison of different training methods**   Naturally, we would like to know how to train a neural network which is certifiably robust against geometric transformations. In this experiment, we evaluate effectiveness of a wide range of training methods to train a network certifiably robust to the translation up to 2 pixels in both $x$ and $y$ direction. While [2] train with adversarial examples and show that this leads to lower empirical success of an adversary, we are interested in a different question: can we train neural networks to be provably robust against geometric transformations?

We first train the same MNIST network as before, in a standard fashion, without any kind of defense. As expected, the resulting network shown in the first row of Table 4 is not robust at all – random attack can find many translations which cause misclassification of the network. To alleviate this problem, we incorporate data augmentation into training by randomly translating every image in a batch between -4 and 4 pixels before passing it through the network. As a result, the network is significantly more robust against the attack, however there are still many images that we fail to certify. To make the network more amenable to certification, we consider two additional techniques. They are both based on the observation that convex relaxation of geometric transformations can be viewed as noise in the pixel values. To train a network which is robust to noise we consider adversarial training with projected gradient descent (PGD) [41] and provable defense based on DIFFAI [23]. We also consider combination of these techniques with data augmentation.

Based on the results shown in Table 4, we conclude that training with PGD coupled with data augmentation achieves both highest accuracy and highest number of certified images. Training with DIFFAI significantly increases certification rate for interval bound propagation, but has the drawback of significantly lower accuracy than other methods.

**Evaluation on large networks**   We evaluated whether DEEPG can certify robustness of large CIFAR-10 networks with residual connections. We certified ResNet-Tiny and ResNet-18 from [42] with 312k and 558k neurons, respectively. As certifying these networks is challenging, we consider relatively small rotation between -2 and 2 degrees. As before, we generated constraints using both DEEPG and interval bound propagation. This experiment was performed on a Tesla V100 GPU.

ResNet-Tiny was trained using PGD adversarial training and has standard accuracy 83.8%. Using the constraints from DEEPG, we certify 91.1% of images while interval constraints allow us to certify only 1.3%. Average time for the verifier to certify or report failure is 528 seconds per image.

ResNet-18 was trained using DIFFAI [42] and has standard accuracy 40.2%. In this case, using constraints from DEEPG, the verifier certifies 82.2% of images. Similarly as before (see Table 4), training the network with DIFFAI also enables a high certification rate of 77.8% even using interval constraints. However, the drawback is that this network has low accuracy of only 40.2% compared to ResNet-Tiny trained with PGD which has 83.8% accuracy. On average, the verifier takes 1652 seconds per image. Here, generating the constraints for both networks took 25 seconds on average.

# 6   Conclusion

We introduced a new method for computing (optimal at the limit) linear constraints on geometric image transformations combining Lipschitz optimization and linear programming. We implemented the method in a system called DEEPG and showed that it leads to significantly better certification precision in proving robustness against geometric perturbations than prior work, on both defended and undefended networks.

**Acknowledgments**

We would like to thank anonymous reviewers for their feedback and Christoph Müller for his help with enabling certification of large residual networks using DeepPoly.

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
