[Supplementary Material]

# A  Additional details of the method

## A.1  Image transformations and their gradients

In this section we give an overview over the transformations used in this paper. For each of the transformations, we list parametrized form of the transformation, then Jacobian of the transformation function (with respect to both inputs and parameters) and finally the inverse function of the transformation.

**Spatial transformations**  Here we list the equations for spatial transformations.

Rotation:

$$R_\phi(x, y) = \begin{pmatrix} \cos\phi & -\sin\phi \\ \sin\phi & \cos\phi \end{pmatrix} \begin{pmatrix} x \\ y \end{pmatrix}$$

$$\partial_{x,y} R_\phi(x, y) = \begin{pmatrix} \cos\phi & \sin\phi \\ -\sin\phi & \cos\phi \end{pmatrix}$$

$$\partial_\phi R_\phi(x, y) = \begin{pmatrix} -x\sin\phi - y\cos\phi \\ x\cos\phi - y\sin\phi \end{pmatrix}$$

$$R_\phi^{-1}(x, y) = \begin{pmatrix} \cos\phi & \sin\phi \\ -\sin\phi & \cos\phi \end{pmatrix} \begin{pmatrix} x \\ y \end{pmatrix}$$

Translation:

$$T_{v_1,v_2}(x, y) = \begin{pmatrix} x + v_1 \\ y + v_2 \end{pmatrix}$$

$$\partial_{x,y} T_{v_1,v_2}(x, y) = \begin{pmatrix} 1 & 0 \\ 0 & 1 \end{pmatrix}$$

$$\partial_{v_1,v_2} T_{v_1,v_2}(x, y) = \begin{pmatrix} 1 & 0 \\ 0 & 1 \end{pmatrix}$$

$$T_{v_1,v_2}^{-1}(x, y) = \begin{pmatrix} x - v_1 \\ y - v_2 \end{pmatrix}$$

Scaling:

$$\mathrm{Sc}_{\lambda_1,\lambda_2}(x, y) = \begin{pmatrix} \lambda_1 & 0 \\ 0 & \lambda_2 \end{pmatrix} \begin{pmatrix} x \\ y \end{pmatrix}$$

$$\partial_{x,y} \mathrm{Sc}_{\lambda_1,\lambda_2}(x, y) = \begin{pmatrix} \lambda_1 & 0 \\ 0 & \lambda_2 \end{pmatrix}$$

$$\partial_{\lambda_1,\lambda_2} \mathrm{Sc}_{\lambda_1,\lambda_2}(x, y) = \begin{pmatrix} x & 0 \\ 0 & y \end{pmatrix}$$

$$\mathrm{Sc}_{\lambda_1,\lambda_2}^{-1}(x, y) = \begin{pmatrix} \frac{1}{\lambda_1} & 0 \\ 0 & \frac{1}{\lambda_2} \end{pmatrix} \begin{pmatrix} x \\ y \end{pmatrix}$$

Shearing:

$$\mathrm{Sh}_m = \begin{pmatrix} 1 & m \\ 0 & 1 \end{pmatrix} \begin{pmatrix} x \\ y \end{pmatrix}$$

$$\partial_{x,y} \mathrm{Sh}_m(x, y) = \begin{pmatrix} 1 & m \\ 0 & 1 \end{pmatrix}$$

$$\partial_m \mathrm{Sh}_m(x, y) = \begin{pmatrix} y \\ 0 \end{pmatrix}$$

$$\mathrm{Sh}_m^{-1} = \begin{pmatrix} 1 & -m \\ 0 & 1 \end{pmatrix} \begin{pmatrix} x \\ y \end{pmatrix}$$

**Interpolation**

$$I^{i,j}(x,y) = \frac{1}{4} \sum_{\substack{v \in \{i, i+2\} \\ w \in \{j, j+2\}}} p_{v,w}(2 - |v - x|)(2 - |w - y|)$$

$$I(x,y) = \begin{cases} I^{i,j}(x,y) \text{ if } (x,y) \in D_{i,j}. \end{cases}$$

$$\partial_x I^{i,j}(x,y) = \frac{p_{i+2,j} - p_{i,j}}{2} + \frac{(y - \hat{y})}{4} A_{i,j}$$

$$\partial_y I^{i,j}(x,y) = \frac{p_{i,j+2} - p_{i,j}}{2} + \frac{(x - \hat{x})}{4} A_{i,j}$$

where $A_{i,j} = p_{i,j} - p_{i,j+2} - p_{i+2,j} + p_{i+2,j+2}$. Furthermore, the bilinear interpolation $I$ is a polynomial of degree 2, which can be seen by performing the rewrite:

$$I^{i,j}(x,y) = \tfrac{p_{i,j}}{4}(i+2-x)(j+2-y) + \tfrac{p_{i+2,j}}{4}(x-i)(j+2-y)$$
$$+ \tfrac{p_{i+2,j}}{4}(i+2-x)(y-j) + \tfrac{p_{i+2,j+2}}{4}(x-i)(y-j)$$

**Brightness and contrast**

$$\mathcal{P}_{\alpha,\beta}(v) = \alpha v + \beta$$
$$\partial_v \mathcal{P}_{\alpha,\beta}(v) = \alpha$$
$$\nabla_{\alpha,\beta}\mathcal{P}_{\alpha,\beta}(v) = (v, 0)^T$$

### A.2 Computing an upper bound on the gradient

Here we explain how to compute an upper bound on the true gradient of the transformation. For the Lipschitz optimization in our running example, we want to calculate the upper bound on the gradient for $I \circ R_\phi(5, 1) - a'_l - c'_l \phi$ on the interval $\phi \in [0, \frac{\pi}{4}]$. Calculating this upper bound amounts to interval propagation on the analytic gradient:

$$(\partial_1 I, \partial_2 I)|_D \cdot (\partial_\phi R_\phi)|_{\phi'} - c'_l$$

The abstraction for the area $D$ is

$$D = R_{[0, \frac{\pi}{4}]}(5, 1) = \begin{pmatrix} [2\sqrt{2}, 5] \\ [\frac{1}{\sqrt{2}}, 1 + \frac{5}{\sqrt{2}}] \end{pmatrix}.$$

We need to calculate the interval of partial derivatives of $I$ for every $I^{i,j}$ separately and join the intervals in the end. The area $D$ has non empty intersection with $\{A_{i,j}\}$ where $(i,j) \in \{1,3\} \times \{-1,1,3\}$. The estimate on $M_{3,1} = D \cap A_{3,1}$ using the concrete pixel values (Fig. 5e), the incline of the linear constraint $w'_l = -0.90$ (§4) and $\phi \in [0, \frac{\pi}{4}]$ we get

$$([0.23, 0.47], [-0.83, -0.24]) \cdot \begin{pmatrix} [-4.53, -0.71] \\ [2.82, 5] \end{pmatrix} - 0.90$$
$$= [-7.18, -1.74].$$

Similar computation can be performed for the other interpolation areas. The resulting intervals from all interpolation areas are joined to make up the final result.

### A.3 Proof of theorem 1

*Proof.* Notice that $L_n$ is essentially a pointwise Monte Carlo estimate of the integral in $L$. Thus, For a fixed $(\boldsymbol{w}, b) \in W$ from the law of large numbers we know that for $\epsilon > 0$, $\Pr(|L_N(\boldsymbol{w}, b) - L(\boldsymbol{w}, b)| < \epsilon)$ tends to 0 as $N$ tends to infinity.

Next, consider a finite number of pairs $(\boldsymbol{w_1}, b_1), \ldots, (\boldsymbol{w_k}, b_k)$. One could apply union bound on the above observation and show that $\Pr(\exists i; |L_N(\boldsymbol{w}_i, b_i) - L(\boldsymbol{w}_i, b_i)| > \epsilon)$ tends to 0 as $N$ tends to infinity.

Finally, one could choose equidistant subdivision of $W$ such that along each dimension of $W$ we make cuts of width $\delta$. Using this subdivision we obtain finite number of $(\boldsymbol{w_1}, b_1), ..., (\boldsymbol{w_k}, b_k)$. Note that for each $(\boldsymbol{w}, b) \in W$ there exists $i$ such that $|\boldsymbol{w} - \boldsymbol{w_i}| < \delta$ and $|b - b_i| < \delta$. Then, from the definition of $L_n$ we can obtain, using triangle and Cauchy-Schwarz inequality:

$$|L_n(\boldsymbol{w}, b) - L_n(\boldsymbol{w_i}, b_i)| = |(b_i - b) - (\boldsymbol{w_i} - \boldsymbol{w})^T \frac{1}{N} \sum_{j=1}^{N} \boldsymbol{\kappa}_j|$$

$$< |(b_i - b)| + |(\boldsymbol{w_i} - \boldsymbol{w})|_\infty |\frac{1}{N} \sum_{j=1}^{N} \boldsymbol{\kappa}_j|_1$$

$$< \delta + \delta \cdot R = \delta(1 + R)$$

Here we denote by $R$ maximum $L_1$ norm of the element in the parameter space (which exists because parameter space is bounded). Analogously, one can obtain that: $|L(\boldsymbol{w}, b) - L(\boldsymbol{w_i}, b_i)| < \delta(1 + R)$. One could choose width $\delta$ small enough such that $\delta(1 + R) < \epsilon$, i.e., $\delta < \frac{\epsilon}{1+R}$.

First, notice that $L_N$ is a pointwise Monte Carlo estimate of the integral in $L$. Thus, due to the law of large numbers for a fixed $(\boldsymbol{w}, b) \in W$ and any $\epsilon > 0$, $\Pr(|L_N(\boldsymbol{w}, b) - L(\boldsymbol{w}, b)| \geq \epsilon)$ tends to 0 as $N$ tends to infinity. Next, consider a finite number of weight-bias pairs $(\boldsymbol{w_1}, b_1), \dots, (\boldsymbol{w_k}, b_k) \in W$. Using union bound on the above probability we know that:

$$\Pr(\exists i \in \{1, 2, ..., k\}, |L_N(\boldsymbol{w_i}, b_i)| - L(\boldsymbol{w_i}, b_i)| > \epsilon) \leq \sum_{j=1}^{k} \Pr(|L_N(\boldsymbol{w_j}, b_j)| - L(\boldsymbol{w_j}, b_j)| > \epsilon).$$

As each summand on the right goes to 0 as we increase $N$ to the infinity, we can conclude that with high probability our estimate $L_N$ is $\epsilon$-close to true function $L$ on all of the $k$ weight-bias pairs.

Finally, we want to prove that our estimate $L_N(\boldsymbol{w}, b)$ is $\epsilon$-close to the function $L(\boldsymbol{w}, b)$, with high probability (so far we have proved this only for finitely many $(\boldsymbol{w}, b)$). In order to show this, we choose equidistant subdivision of $W$ such that along each dimension of $W$ we make cuts of width $\delta$. Using this subdivision we obtain finite number of $(\boldsymbol{w_1}, b_1), ..., (\boldsymbol{w_k}, b_k)$. Note that for each $(\boldsymbol{w}, b) \in W$ there exists $i$ such that $|\boldsymbol{w} - \boldsymbol{w_i}| < \delta$ and $|b - b_i| < \delta$. Then, from the definition of $L_N$ we can obtain, using triangle and Cauchy-Schwarz inequality:

$$|L_N(\boldsymbol{w}, b) - L_N(\boldsymbol{w_i}, b_i)| = |(b_i - b) - (\boldsymbol{w_i} - \boldsymbol{w})^T \frac{1}{N} \sum_{j=1}^{N} \boldsymbol{\kappa}_j|$$

$$< |(b_i - b)| + |(\boldsymbol{w_i} - \boldsymbol{w})|_\infty |\frac{1}{N} \sum_{j=1}^{N} \boldsymbol{\kappa}_j|_1$$

$$< \delta + \delta \cdot R$$
$$= \delta(1 + R)$$

Here $R$ denotes the maximum $L_1$ norm of the element in the parameter space (which exists because parameter space is bounded). Analogously, one can obtain that: $|L(\boldsymbol{w}, b) - L(\boldsymbol{w_i}, b_i)| < \delta(1 + R)$. We can choose width $\delta$ small enough such that $\delta(1 + R) < \epsilon$, i.e., $\delta < \frac{\epsilon}{1+R}$.

Then:

$$|L_N(\boldsymbol{w}, b) - L(\boldsymbol{w}, b)|$$
$$\leq |L_N(\boldsymbol{w}, b) - L_n(\boldsymbol{w_i}, b_i)| + |L_N(\boldsymbol{w_i}, b_i) - L(\boldsymbol{w_i}, b_i)| + |L(\boldsymbol{w_i}, b_i) - L(\boldsymbol{w}, b)|$$
$$< \epsilon + \epsilon + \epsilon$$
$$< 3\epsilon.$$

Now, we know that for finite subdivision of weights our estimate $L_n$ is $\epsilon$-close to $L$, with high probability. From the above, we get that if our subdivision is fine enough ($\delta$ small enough) then $L_n$ is $\epsilon$-close to $L$ on the entire weights space $W$. This way we obtain that:

$$\Pr(\exists (w, b) \in W; |L_N(\boldsymbol{w}, b) - L(\boldsymbol{w}, b)| > \epsilon) \overset{N \to \infty}{\Rightarrow} 0.$$

Now we obtained that for $N$ large enough $L_N$ is $\epsilon$-close to $L$ pointwise, with high probability. Finally, let $(\boldsymbol{w}', b')$ and $(\boldsymbol{w}^*, b^*)$ be minimums of $L_N$ and $L$, respectively. We find that:

$$
\begin{aligned}
&L(\boldsymbol{w}', b') - L(\boldsymbol{w}^*, b^*) \\
&= (L(\boldsymbol{w}', b') - L_N(\boldsymbol{w}', b')) + (L_N(\boldsymbol{w}', b') - L_N(\boldsymbol{w}^*, b^*)) + (L_N(\boldsymbol{w}^*, b^*) - L(\boldsymbol{w}^*, b^*)) \\
&< \epsilon + 0 + \epsilon \\
&= 2\epsilon.
\end{aligned}
$$

$\square$

### A.4 Computation of interval pixel bounds

This section generalizes [9] to arbitrary combinations of simple geometric transformations. To compute interval bounds for each pixel in the transformed image with respect to a hyperrectangle of parameters

$$(\alpha, \beta, \boldsymbol{\mu}) \in h := [a_1, b_1] \times \cdots \times [a_n, b_n] \subset \mathbb{R}^n,$$

the algorithm first partitions the parameter space into $s$ *splits* $\{h_k\}_{k \in [s]}$. Then, for each split $h_k$, it computes the interval bounds for every pixel value $\tilde{p}_{x,y}$ as follows (($x, y$) fixed):

**Step 1** Partition the current split $h_k$ into $r$ *refined splits* $\{h_{kl}\}_{l \in [r]}$ so to increase accuracy of approximation.

**Step 2** For every $l \in [r]$, calculate a pair of intervals $D^{(l)} = [d_1, d_1'] \times [d_2, d_2'] \subset \mathbb{R}^2$ which approximate the reachable coordinates induced by $\{\mathcal{T}_{\boldsymbol{\mu}}^{-1}(x, y) \mid \forall (\alpha, \beta, \boldsymbol{\mu}) \in h_{kl}\}$. To accomplish this, we use the fact we can easily obtain the closed form of the inverse of the transformations we consider (Appendix A.1). For example, for rotations, $R_{-\phi} = R_{\phi}^{-1}$. We can then propagate the interval bounds through this inverse.

**Step 3** Calculate the interval approximation $\iota_{kl}$ for every $D^{(l)}$. This is done by propagating the pair of intervals $D^{(l)}$ through $\mathcal{P}_{[a_1, b_1], [a_2, b_2]} \circ I$, that is

$$
\begin{aligned}
\iota_{kl} &= \mathcal{P}_{[a_1, b_1], [a_2, b_2]} \circ I(D^{(l)}) \\
&= \mathcal{P}_{[a_1, b_1], [a_2, b_2]} \left( \bigcup_{i,j} I^{i,j}(M_{i,j}^{(l)}) \right).
\end{aligned}
$$

Here we have that $M_{i,j}^{(l)} := D^{(l)} \cap A_{i,j}$, that is, we intersect the approximation of the reachable coordinates $D^{(l)}$ with the interpolation region $A_{i,j}$.

| | 5 | 0 | 0 | 0.35 |
|---|---|---|---|---|
| | 3 | 0.36 | 0.83 | 0.97 |
| | 1 | 0.99 | 0.99 | 0.99 |
| | -1 | 0.99 | 0.99 | 0.99 |
| | | 1 | 3 | 5 |

(a) Original image  (b) Interpolation  (c) Intersected regions  (d) Rotated image

(e) Original pixel values

Figure 5: Image rotation by $-\frac{\pi}{4}$ degrees. Here, (a) shows the original image, while (b) and (c) show an excerpt of (a) focusing on particular interpolation regions. They also include regions computed by the interval analysis. Finally, (d) shows the transformed rotated image.

**Step 4**  Finally, we combine (join) all interval approximations $\iota_{kl}$ into a single interval $\iota_k = \cup_{l \in [r]} \iota_{kl}$, capturing the possible values $\tilde{p}_{x,y} = \mathcal{I}_{\alpha,\beta,\boldsymbol{\mu}}(x,y)$ that pixel $(x,y)$ can take on for the split $h_k$.

As we show later, the interval range $\iota_k$ computed for each of the resulting splits $h_k$ can then be used for certification or for adversarial example generation.

**Running example**  To illustrate the above steps on our example, consider again pixel $(5,1)$ and the angle range $\phi \in [-\frac{\pi}{2}, 0]$. This range corresponds, using $R_{-\phi} = R_\phi^{-1}$, to $\phi \in [0, \frac{\pi}{2}]$. For $s = 2$, we obtain splits $[0, \frac{\pi}{4}]$ (the small red arch in Fig. 2a which is zoomed in in Fig. 2b) and $[\frac{\pi}{4}, \frac{\pi}{2}]$. As discussed above, the interval approximations are produced independently for both splits. Let us consider the split $[0, \frac{\pi}{4}]$ and refine it by splitting it again in half ($r = 2$). The approximation $D^{(1)}$ for the first refined split $h_{11} := [0, \frac{\pi}{8}]$ is

$$D^{(1)} = R_{[0,\pi/8]}(5,1) = \begin{pmatrix} \cos(h_{11}) & -\sin(h_{11}) \\ \sin(h_{11}) & \cos(h_{11}) \end{pmatrix} \begin{pmatrix} 5 \\ 1 \end{pmatrix}$$

$$= \begin{pmatrix} [\cos\frac{\pi}{8}, 1] & [-\sin\frac{\pi}{8}, 0] \\ [0, \sin\frac{\pi}{8}] & [\cos\frac{\pi}{8}, 1] \end{pmatrix} \begin{pmatrix} 5 \\ 1 \end{pmatrix} = \begin{pmatrix} [5\cos\frac{\pi}{8} - \sin\frac{\pi}{8}, 5] \\ [\cos\frac{\pi}{8}, 1 + 5\sin\frac{\pi}{8}] \end{pmatrix}.$$

Similarly, one obtains $D^{(2)}$. The green rectangles shown in Fig. 2b capture the values of $D^{(1)}$ and $D^{(2)}$. Here, $D^{(1)}$ intersects the interpolation regions $A_{3,-1}$ and $A_{3,1}$, shown in Fig. 5c. The interval approximations for the values on $M_{3,-1}^{(1)} = D^{(1)} \cap A_{3,-1}$ using Eq. (1) are

$$I^{3,-1}\left(M_{3,-1}^{(1)}\right) = I([5\cos\frac{\pi}{8} - \sin\frac{\pi}{8}, 5], [\cos\frac{\pi}{8}, 1])$$
$$\approx [0.58, 1],$$

where the upper bound of the interval was cut back to 1. Similarly, we obtain $I^{3,1}\left(M_{3,1}^{(1)}\right) \approx [0.02, 1]$. Thus, the interval approximation for pixel $(5,1)$ on the first refinement split $\phi \in [0, \frac{\pi}{8}]$ is $[0.58, 1] \cup [0.02, 1] = [0.02, 1]$.

The approximation of the second refined split $D^{(2)}$ intersects four interpolation regions: $A_{1,1}$, $A_{1,3}$, $A_{3,1}$ and $A_{3,3}$. This intersection is also shown in Fig. 5c. Via similar calculations as above we obtain

$$I^{1,1}\left(M_{1,1}^{(2)}\right) \approx [0.61, 1], \quad I^{1,3}\left(M_{1,3}^{(2)}\right) \approx [0.20, 0.87],$$
$$I^{3,1}\left(M_{3,1}^{(2)}\right) \approx [0.25, 1], \quad I^{3,3}\left(M_{3,3}^{(2)}\right) \approx [0.08, 1].$$

Thus, the interval approximation for the second refinement split $\phi \in [\frac{\pi}{8}, \frac{\pi}{4}]$ is $[0.08, 1]$.

In the final step 4, the two intervals $[0.08, 1]$ and $[0.02, 1]$ are combined to obtain the range $[0.02, 1]$ for the split $[0, \frac{\pi}{4}]$ corresponding to $0.02 \leq \mathcal{I}_{[-\frac{\pi}{4}, 0]}(5, 1) \leq 1$.

In the next section, we show how to improve the precision of the interval overapproximation $[c_l, c_u]$, corresponding to $c_l \leq \mathcal{I}_{\alpha,\beta,\boldsymbol{\mu}}(x,y) \leq c_u$, by using linear upper and lower constraints to capture the correlation between pixel values and parameters $(\alpha, \beta, \boldsymbol{\mu})$.

### A.5 Lipschitz optimization with bound refinement

Here we provide more details of our algorithm for Lipschitz optimization. In order to obtain a sound linear constraint, we need to find sound overapproximation of maximum of the function $f(\boldsymbol{\kappa}) = b'_l + \boldsymbol{w}_l'^T \boldsymbol{\kappa} - \mathcal{I}_{\boldsymbol{\kappa}}(x,y)$ on a hyperrectangle domain. The result is an interval $[f_{\max}, f_{\max} + \epsilon]$ which contains true maximal value $f^\star$ of function $f$.

**Lipschitz continuity**  A function $f \colon D \subset \mathbb{R}^n \to \mathbb{R}$ is Lipschitz continuous with a Lipschitz constant $L \in \mathbb{R}_{\geq 0}$ if $\|f(\boldsymbol{y}) - f(\boldsymbol{x})\| \leq L \|\boldsymbol{y} - \boldsymbol{x}\|$ for all $\boldsymbol{x}, \boldsymbol{y} \in D$.

If $D$ is compact, the Lipschitz constant of a smooth function $f$ is equal to $\max_{\boldsymbol{x} \in D} \|\nabla f|_{\boldsymbol{x}}\|$. We also use this equation for the (non-smooth) bilinear interpolation $I$. As mentioned in Section 3, $I$ is smooth on every interior of $A_{i,j}$. Further, $I|_{A_{i,j}}$ can be extended smoothly at the boundary of $A_{i,j}$ by $I^{i,j}$. This allows us to approximate the Lipschitz constant $L$ of $I$ over some compact region $D \subset \mathbb{R}^2$,

using the approximated gradient $\nabla^D I = \bigcup_{i,j} \nabla I^{i,j}(D \cap A_{i,j})$, by $\|\nabla^D I\|$, where the approximation can be computed using any overapproximation based analysis (we use intervals, but one can plug in other approximations such as zonotope).

**Lipschitz optimization with iteratively refined bounds**   The objective we aim to solve is finding the $\epsilon$ optimal value $f_{\max}$ of the Lipschitz continuous function $f$ over a hyperrectangle $h = \prod_{i=1}^{n} [a_i, b_i] \subset \mathbb{R}^n$.

To address this problem, we adapt a branch-and-bound algorithm for solving constrained (multidimensional) optimization tasks [38] to piecewise differentiable Lipschitz continuous functions. In particular, we show how to leverage any overapproximation based analysis (e.g., interval analysis) to soundly approximate local gradients (Lipschitz constants). This process is repeated at every step, thus obtaining more refined bounds for the local gradients at every iteration. Overall, this method leads to significantly fewer iterations of the optimization algorithm, improves convergence speed at flat areas, and removes the need to know the global Lipschitz constant in advance.

The inputs to the algorithm (Algorithm 1) are a Lipschitz continuous function $f$, the hyperrectangle $h$ over which we optimize and the number of cuts $k \in \mathbb{N}_{\geq 2}$ in which a hyperrectangle will be split to gain precision. The idea is to constrain the range of the true maximum value $f^\star$ by increasing its lower bound $f_{\max}$ (lower bound on all of $h$) while decreasing all of its upper bounds $f_{h'}^{\text{bound}}$ on the subrectangles $h'$ that need to be considered.

We use a priority queue $q$ to store pairs of the form $(h', f_{h'}^{\text{bound}})$, where $h'$ corresponds to a subrectangle of $h$, potentially containing the true maximal value $f^\star$, and $f_{h'}^{\text{bound}}$ corresponds to an upper bound (Eq. (9) or Eq. (10)) of $f$ in $h'$. The queue is ordered by the upper bounds. To improve our bounds on $f^\star$, we pop the top of the queue $(h', f_{h'}^{\text{bound}})$ and split $h'$ into $k$ splits $h'_1, \dots, h'_k$ according to Eq. (11) or Eq. (12). The lower bound $f_{\max}$ is updated to $f$ evaluated at a center of $h'_i$, if this would increase $f_{\max}$. The new pairs $(h'_i, f_{h'_i}^{\text{bound}})$ are added to the queue as long as the new bound $f_{h'_i}^{\text{bound}}$ is larger than $f_{\max} + \epsilon$, and thus may help to increase $f_{\max}$ by more than $\epsilon$. The algorithm terminates if the largest upper bound $f_{h'_i}^{\text{bound}}$ and the lower bound $f_{\max}$ are $\epsilon$-close.

**Bounds**   The function $f$ on the hyperrectangle $h' = [a'_1, b'_1] \times \cdots \times [a'_n, b'_n]$ can be bounded using the (commonly used) Cauchy-Schwarz bound or the triangle bound

$$f_h^{\text{cs-bound}}(h, \nabla f) = f(c_h) + \frac{L}{2} \sqrt{\sum_{i=1}^{n} (b_i - a_i)^2} \tag{9}$$

$$f_h^{\text{t-bound}}(h, \nabla f) = f(c_h) + \frac{1}{2} \sum_{i=1}^{n} v_i (b_i - a_i) \tag{10}$$

where we compute the approximation (e.g., interval or zonotope) of the gradient $\nabla^h f$ on $h$ to calculate $L$ as the upper bound of the approximation $\|\nabla^h f\|$ and $v_i$ is the upper bound of the approximation $\|\partial_i^h f\|$. We use the triangle bound as it is tighter.

**Triangle bound**   Let $f \colon D \to \mathbb{R}$ be a smooth function and $D$ be a convex set. Using the mean value theorem, we know that for all $\boldsymbol{x}, \boldsymbol{y} \in D$ there exists a $\boldsymbol{z} \in D$ such that

$$f(\boldsymbol{y}) - f(\boldsymbol{x}) = \nabla f(\boldsymbol{z})^T (\boldsymbol{x} - \boldsymbol{y}) = \sum_i \partial_i f(z)(\boldsymbol{x}^{(i)} - \boldsymbol{y}^{(i)})$$

$$\leq \sum_i \max_{\boldsymbol{z} \in D} |\partial_i f(z)| |\boldsymbol{x}^{(i)} - \boldsymbol{y}^{(i)}|$$

which results into

$$f(\boldsymbol{y}) \leq f(\boldsymbol{x}) + \sum_i \max_{\boldsymbol{z} \in D} |\partial_i f(z)| |\boldsymbol{x}^{(i)} - \boldsymbol{y}^{(i)}|.$$

We note, that in our case $f = \mathcal{I}$ as in Section 3, the inequality still holds: The points where $f$ is not differentiable can be handled by applying the inequality piecewise.

Table 5: Hyperparameters used in our experiments. Splits denotes number of domain splits per dimension (same for Interval and DEEPG). $k$ is number of refinements per dimension used for Interval baseline. Last two columns show number of samples $n$ used to solve LP and $\epsilon$ in Lipschitz optimization in DEEPG.

|  |  | Splits | $k$ (Interval) | $n$ (DEEPG) | $\epsilon$ (DEEPG) |
|---|---|---|---|---|---|
| MNIST | R(30°) | 10 | 10 000 | 1 000 | 0.0001 |
|  | T(2,2) | 11 | 150 | 2 000 | 0.00001 |
|  | Sh(2), R(2°), Sc(2), B(2, 0.001) | 1 | 12 | 1 000 | 0.006 |
| FashionMNIST | Sc(20) | 10 | 10 000 | 1 000 | 0.0001 |
|  | R(10), B(2, 0.01) | 4 | 25 | 2 000 | 0.002 |
|  | Sc(3), R(3), Sh(2) | 2 | 35 | 1 000 | 0.001 |
| CIFAR-10 | R(10°) | 20 | 10 000 | 1 000 | 0.00001 |
|  | R(2), Sh(2) | 2 | 50 | 1 000 | 0.0001 |
|  | Sc(1), R(1°), B(1, 0.001) | 2 | 12 | 1 000 | 0.001 |

**Partitioning** The hyperrectangle $h$ can be split into $k$ hyperrectangles of equal size by cutting one of its edges $l$ into $k$ equal parts. The common choice for $l$ is to use the longest edge. We refine this by weighing the edge length with gradient information

$$l = \arg\max_l (b'_l - a'_l) \qquad \text{largest edge} \tag{11}$$

$$l = \arg\max_l v_l(b'_l - a'_l) \quad \text{largest weighted edge} \tag{12}$$

# B    Additional details for experimental evaluation

In this section we list additional details which are necessary to reproduce our experimental results.

## B.1    Polyhedra transformers

Here we describe Polyhedra transformers used in our experiments. Note that all geometric transformations are linear which is why polyhedra approximation for the transformation itself does not lose precision. After the transformation is applied, we obtain exact linear expression for new position of every pixel. Next operation in the sequence is bilinear interpolation which is polynomial of degree 2 and involves multiplication between two linear expressions (addition which is part of the polynomial is again exact in polyhedra). Multiplication cannot be captured exactly and we need to decide how to lose precision. We choose standard solution of concretizing one of the polyhedra expressions to interval (we concretize one which produces interval with smaller width).

Finally, we need to join polyhedra constraints over all interpolation regions. This amounts to computing lower and upper convex hull of polyhedra constraints. This is challenging problem, however for the case of 1-dimensional transformations there is relatively easy solution. If parameter of the transformation is in the interval $[l, u]$ we intersect polyhedra constraints with lines $y = l$ and $y = u$ and choose smallest (largest) intercept with each of the vertical lines and join the intercepts, thus forming a new linear lower bound.

## B.2    Parameter configurations for experiments

Table 7 shows architectures used for our experiments on MNIST, Fashion-MNIST and CIFAR-10 datasets. In Table 5 we show parameter configurations for all of our experiments. As noted before, parameters are chosen in such a way that runtime of both DEEPG and interval analysis are roughly the same. We did not observe a significant difference in performance when changing the parameters. We report runtime for our experiments in Table 6.

**MNIST** In the experiments on MNIST, in all of the training methods we consider, the network is trained for 100 epochs using batch size 128. We use Adam optimizer and with initial learning rate 0.001 which we decay by 0.5 every 10 epochs. For the experiments which use PGD training, we

Table 6: For every experiment, breakdown of runtime between time taken to compute the constraints using DEEPG and time taken to certify the network using DeepPoly.

|  |  | Runtime (seconds) | |
|  |  | Constraint generation (DEEPG) | Certification (DeepPoly) |
| --- | --- | --- | --- |
| MNIST | R(30) | 10 | 25 |
|  | T(2, 2) | 206 | 57 |
|  | Sc(5), R(5), B(5, 0.01) | 162 | 14 |
|  | Sh(2), R(2), Sc(2), B(2, 0.001) | 50 | 1 |
| Fashion-MNIST | Sc(20) | 9 | 6 |
|  | R(10), B(2, 0.01) | 174 | 47 |
|  | Sc(3), R(3), Sh(2) | 95 | 6 |
| CIFAR-10 | R(10) | 49 | 68 |
|  | R(2), Sh(2) | 13 | 14 |
|  | Sc(1), R(1), B(1, 0.001) | 179 | 60 |

Table 7: Architectures used in experimental evaluation.

| MNIST | FashionMNIST | CIFAR-10 |
| --- | --- | --- |
| CONV 32 4×4 + 2 | CONV 32 4×4 + 1 | CONV 32 4×4 + 1 |
| CONV 64 4×4 + 2 | CONV 32 4×4 + 2 | CONV 32 4×4 + 2 |
| FC 200 | CONV 64 4×4 + 2 | CONV 64 4×4 + 2 |
| FC 10 | FC 150 | FC 150 |
|  | FC 10 | FC 10 |

train with $\epsilon = 0.1$ and perform 40 steps with the step size 0.005 in every iteration. We also use $L_1$ regularization factor of 0.00005. For the experiments with FashionMNIST we use exactly the same setup.

**CIFAR-10**    In the experiments on CIFAR-10, in all of the training methods we consider the network is trained for 100 epochs using batch size 128. We use SGD optimizer and with initial learning rate 0.01 which we decay by 0.5 every 10 epochs. For the experiments which use PGD training, we train with $\epsilon = 0.005$ and perform 7 steps with the step size 0.002 in every iteration. We also use $L_1$ regularization factor of 0.00001, except for the experiment with rotation and shearing where we use 0.00005.