[Reviews · NeurIPS 2019]

Reviewer 1



This work borrows from recent research on certifying robustness properties of neural networks on image datasets to Linfinity norm perturbations, and extends this to certification against geometric attacks. More specifically, previous works used sound approximations on the outputs of non-linear activations to certify against worst-case Linfinity based perturbations. Accumulating these approximations gives a (potentially loose) output region that can be certified as robust. This work extends and develops techniques to handle geometric based attacks such as rotation and scaling. Overall, I found this paper to be well written. I particularly appreciated the running example style employed in this paper for exposition. I list my suggestions and concerns below: 1. Why is the input always assumed to be perturbed with Linfinity noise? There is no justification for this in the text as far as I can see, its inclusion is unnecessary and somewhat confusing given the main contribution is to defend against other kinds of perturbations. 2. It is surprising that the networks tested have non-negligible robustness to geometric attacks (even the undefended networks). Engstrom et al [1] (and others) have shown that simple attacks such rotating the input usually causes a misclassification, why are the verified networks here seemingly robust to these attacks? 3. The networks verified are extremely small. How difficult would it be to scale to larger datasets / networks? It seems that due to the optimization & sampling mechanism employed this would suffer more than related work (such as IBP [2] which only requires two passes through the network to propagate linear bounds). For example, to reduce sampling error to <= 0.5% takes over 45s - this seems like it has a scalability issue. 4. Adversarial attacks are commonly restricted to small epsilon values for the Linfinity norm since this is highly likely to preserve semantic meaning of the input. Where in the paper do you define a similarity metric under geometric attacks? For example, an a horizontally flipped two digit can be interpreted as a five - yet your certification scheme would not exclude such cases. [1] Engstrom et al. A Rotation and a Translation Suffice: Fooling CNNs with Simple Transformations [2] Gowal et al. On the Effectiveness of Interval Bound Propagation for Training Verifiably Robust Models

Reviewer 2



Significance: The field of adversarial robustness and verification currently focuses mostly on various L_p metrics which are not aligned with human perception (e.g. a shift of an image does not change the content but has high L_p differenc). Here the authors take various geometric transformations and provide certifications for the robustness. Style: Well written and clear structure and intuitive explanation with a running example. Questions: * How does this relate to vector field based transformations? Is it a subset thereof?

Reviewer 3



Originality: The method is definitely new and previous work that attempted to do verification on rotation attacks is discussed. Quality: -> The method described is technically sound. The only caveat that I have is with regards to the scalability. Unless I'm mistaken, this requires to, for each split of the hyperparameter, for each pixel, solve a LP using Gurobi to obtain the linear bound candidates, and then perform the Branch and Bound lipshitz optimization to get the offset, which seems extremely expensive. What do the runtime in Table 3 correspond to? Mean verification time for an image? Just the time to derive the linear bounds? Time to derive the linear bounds for 1 pixel? -> With regards to evaluating the performance, the method compares very favourably to previous techniques (interval analysis) which are pretty inadapted to the task. This shows why the method proposed is interesting but it would also be beneficial to get an estimate of the limitation of the method. One ideal way would be to get some upper bounds on the verifiability that can be achieved, maybe as the result from some geometrical adversarial attacks? Clarity: The methods is described sufficiently well that it would be possible for a motivated reader to reproduce the results. Significance: The improvement proposed in this paper is of interest because while a significant amount of progress was made in bounding the outputs of Neural Networks given initial constraints, there hasn't been as much work in describing how to encode richer properties beyond linear constraints or norm-limited perturbations. Also, some more local comments that I had at various places: L.87, paragraph "Geometric Image transformations" I don't understand the motivation for assuming that all the pixels are first perturbed by a small L-inf noise. If the robustness of interest is against rotation, what is the point of including also robustness to small perturbation? L.140, Eq 5, Eq 6. V has not been introduced. L.155 to 159: This paragraph exactly describes the process of Branch-and-Bound, which is extremely standard in optimization. This might be good to at least point the connection for the benefits of the readers. More generally, on the subject of this Branch and Bound procedure, the method proposed here is to run a branch and bound type method to obtain the offset to add to the linear constraint such that it is sound. Once this offset has been computed, it's not entirely sure that the bound on the intensity will be good enough to guarantee robustness. Would there be some benefits in just using a looser bound for the nu_l offset (essentially skipping the branching at that stage to save computation) but instead doing branching at a higher level so as to refine the splits on the possible parameters if the full verification (using DeepPoly) isn't tight enough? This would have the added benefits to make things significantly tighter (imagine in Figure 3 being able to introduce a split at pi/8, this additional split would massively reduce the volume of the overapproximation) l.223. typo "verifer" [Updated review] I read the comments by the other reviewers, as well as the rebuttal by the author. At the moment, my opinions are as follows: -> The scalabilty of the network is in line with usual formal verification of NN work. -> According to their comment (l.37 rebuttal), given enough computational budgets (I hope they would put the time it took but I can imagine a reasonable estimate) they can actually perform verification quite well (the bounds are not ridiculously loose) -> I'm satisfied with the explanation given by the authors with regards to their decision to get as tight as possible bounds before passing along to the verifiers, due to the difference in runtime of the two components of the pipelines. I think that the paper is interesting and in my experience, the problem they tackle is one that regularly comes up in discussion with people interested in formal verification of NN, so there would definitely be interest by the community once published. I'm raising my score as both my question have been adressed

[Author Response · NeurIPS 2019]

We thank the reviewers for their valuable comments which we will incorporate in our work.

**Reviewers #1, #3:** *What is the scalability of your method? How scalable it is to larger datasets / networks?*

First, we remark the time to compute the bounds (our contribution) is not the main bottleneck, but the propagation of these bounds through the network with a state-of-the-art verifier. E.g., bound computation for 1 image and 1 split typically takes few seconds while bound propagation through a moderately sized network takes $\sim 50$ seconds (larger networks increase this time). Switching to other verifiers is unlikely to help as they report similar times [8, 15, 21, 25].

We now elaborate more on the scalability of our method. First, we clarify that *runtime* in Table 3 corresponds to the total time it takes to compute the pixel bounds for 1 image and 1 parameter split, averaged over all splits on the test set. We now also computed the same runtime metric for all experiments in Table 1. The results, in seconds, are:

MNIST: 0.6, 1.8, 11, 36     FashionMNIST: 0.1, 1.4     CIFAR-10: 2.5, 2.5, 1.6, 18

All parameters used for experiments in Table 1 are shown in Table 5. Generally, as also indicated by Table 3, we find that a relatively small number of samples $n$ for which an LP solution is found quickly combined with low $\epsilon$-tolerance so that branch-and-bound terminates quickly (e.g., $n = 100, \epsilon = 0.01$), are sufficient to reach high verified robustness (96.5%) fast (1.2 seconds). Further decrease of $\epsilon$ and increase of $n$ brings small benefits in verified robustness (1.7%).

We also ran our method on ImageNet – the method takes $\sim 2$ minutes per image due to increased number of pixels. The main issue here is that all existing verifiers lose too much precision when propagating constraints through a full blown ImageNet-sized network. Finally, we ran verification of our pixel bounds through a larger network (62K neurons), obtaining similar robustness to the network used in Table 1 (though expectedly, verification time increases).

We note that using IBP for both bound computation and propagation will be more scalable but suffer from very low precision – strictly worse than the Interval baseline of Table 1, which is already much worse than our method.

We will add all updated results and above clarifications to the paper.

**Reviewers #1, #3:** *Why is the input always assumed to be perturbed with $L_\infty$ noise?*

Our method does not assume $L_\infty$ noise and we do have experiments without it (see Table 1). We did perform some experiments with $L_\infty$ noise following Singh et al. [5] who certified the specific composition of $L_\infty$ noise and rotation.

**Reviewer #1:** *Why are the verified networks here seemingly robust to these attacks?*

This is because the networks are trained using standard data augmentation (e.g., if we verify rotations, we augment data with rotations). Note that the same training method is also considered by Engstrom et al. [2] and is shown to significantly increase robustness to geometric transformations compared to networks trained without this augmentation.

**Reviewers #1, #2:** *Do you define a similarity metric under geometric attacks?*

We do not define such a metric in this work but focus on classic transformations (e.g., rotations) which are parameterized. As usual, the user specifies the parameters for which they want to certify the network (e.g., the -30 to 30 degrees for an MNIST image used in Table 1 can be argued to contain images that are indeed visually similar to humans).

**Reviewer #2:** *How does this relate to vector field based transformations? Is it a subset thereof?*

Our transformations capture the most common instances of vector field transformations, but not all. Note that generally vector field transformations are not guaranteed to preserve image similarity (unless bounded by a norm) which is why we focus on transformations known to produce similar images according to human perception (e.g., rotations).

**Reviewer #3:** *Can you provide an upper bound on verifiability?*

Yes. We computed an upper bound for the first experiment on CIFAR-10 with rotations $\in [-10, 10]$. We performed "Worst-of-k" attack from [2] which, for every image, randomly samples 100 parameter choices and checks for misclassifications. This gives an upper bound of 73% (verification rate is 51.5% in Table 1). We also ran DeepG with twice as many parameter splits and verified 72% of images (only 1 image remained). In general, DeepG gets close to the upper bound by increasing the number of splits. However, such increase in the number of splits is only possible if the # of parameters is small (otherwise, the cost is prohibitive). We will add these results to the paper.

**Reviewer #3:** *Can you use a looser offset bound and do branching for parameter refinement at a higher level?*

This is an interesting idea and we considered it earlier. The main problem is that each refinement requires a call to the verifier, which is the main bottleneck as mentioned above. Ideally, there would be a policy (branch and bound or another heuristic) which refines parameters so the number of verifier calls is minimized (so far we did not find a policy which noticeably improves over the uniform refinement used in our work). Importantly, while interesting, this direction is orthogonal to our approach as any split will benefit from more precise linear bounds.

[Meta-Review · NeurIPS 2019]

This is apparently the first paper which can verify that the classification does not change under non-trivial combinations of meaningful geometric transformations like rotations, translations, scaling, shear etc. I think this is an important step in the right direction in the area of provable robustness guarantees for neural networks. For the final version I recommend to - include test errors of all the models - train models with data augmentation and adversarial data augmentation and show how that affects the robustness guarantees. This will increase the practical impact of this paper.